# An approach for developing a blood-based screening panel for lung cancer based on clonal hematopoietic mutations

Ramu Anandakrishnan[ID][1,2]*, Ryan Shahidi[1], Andrew Dai[1], Veneeth Antony[1], Ian J. Zyvoloski[3]

**1** Edward Via College of Osteopathic Medicine, Biomedical Sciences, Blacksburg, Virginia, United States of America, **2** Maryland-Virginia College of Veterinary Medicine, Virginia Tech, Blacksburg, Virginia, United States of America, **3** University of Maryland, Baltimore, Maryland, United States of America

* ramu@vt.edu

**Data Availability Statement:** The datasets used in the current study are available in the Genomic Data Commons Data Portal (https://portal.gdc.cancer.gov/) and the Sequencing Read Archive (https://

## Abstract

Early detection can significantly reduce mortality due to lung cancer. Presented here is an approach for developing a blood-based screening panel based on clonal hematopoietic mutations. Animal model studies suggest that clonal hematopoietic mutations in tumor infiltrating immune cells can modulate cancer progression, representing potential predictive biomarkers. The goal of this study was to determine if the clonal expansion of these mutations in blood samples could predict the occurrence of lung cancer. A set of 98 potentially pathogenic clonal hematopoietic mutations in tumor infiltrating immune cells were identified using sequencing data from lung cancer samples. These mutations were used as predictors to develop a logistic regression machine learning model. The model was tested on sequencing data from a separate set of 578 lung cancer and 545 non-cancer samples from 18 different cohorts. The logistic regression model correctly classified lung cancer and non-cancer blood samples with 94.12% sensitivity (95% Confidence Interval: 92.20–96.04%) and 85.96% specificity (95% Confidence Interval: 82.98–88.95%). Our results suggest that it may be possible to develop an accurate blood-based lung cancer screening panel using this approach. Unlike most other "liquid biopsies" currently under development, the approach presented here is based on standard sequencing protocols and uses a relatively small number of rationally selected mutations as predictors.

## Introduction

Cancer is a leading cause of death in the US, second only to heart disease, with lung cancer accounting for an estimated 21% of these deaths [1]. Early detection of lung cancer has been shown to reduce mortality by 20% [2]. However, the recommended screening for lung cancer, low dose computed tomography [3], is restricted to high-risk individuals, with an estimated uptake of only 17.54% even within this group due to financial and other barriers [4]. Clearly, there is a need for more accessible lung cancer screening protocols. Several blood-based screening panels ("liquid biopsies") are currently in various stages of validation, though none

www.ncbi.nlm.nih.gov/sra) repositories. The accession numbers or identifiers for the datasets are listed in SI Table S1. Source code is available at https://sourceforge.net/projects/lung-cancer-screening-panel/.

**Funding:** This work was funded by the VCOM 2023 REAP grant #1038624(RA). The funders had no role in study design, data collection and analysis, decision to publish, or preparation of the manuscript.

**Competing interests:** The authors have declared that no competing interests exist.

have yet achieved clinical utility [5], other than as companion diagnostics [6]. These include screens for antibodies [7], circulating microRNA [8], and cell-free DNA fragments [9]. For example, Galleri, a multi-cancer blood test based on the methylation state of cell-free DNA, is currently in clinical trials even though it had a poor 51.5% sensitivity (95% confidence interval (CI): 49.6–53.3%), but a high 99.5% specificity (95% CI: 99.0–99.8%) in a clinical validation study [10].

Here we present an approach for developing a blood-based lung cancer screening panel based on a set of clonal hematopoietic (CH) mutations. Clonal hematopoiesis is the clonal expansion of a single hematopoietic stem or progenitor cell (HSPC) due to specific mutations in these cells. While mutations in *tumor cells* that inhibit anti-tumor immune function have been extensively studied [11, 12], the potential impact of <u>*immune cell mutations*</u> on anti-tumor activity remains relatively unexplored [13]. Recent studies have shown that CH mutations can lead to the clonal expansion of potentially pathogenic mutations in a relatively large population of tumor infiltrating immune (TII) cells, affecting their anti-tumor activity. A loss-of-function *TET2* mutation in tumor infiltrating myeloid cells was shown to increase angiogenesis in a lung cancer animal model, exacerbating tumor progression [14]. An *ASXL1* mutation in tumor infiltrating T-cells was shown to perturb their development and function, promoting tumor growth in syngeneic animal models [15]. On the other hand, *TET2* inactivation in tumor infiltrating lymphocytes and myeloid cells has also been shown to inhibit tumor growth [16, 17], indicating a dependence on other factors. Studies have also shown that mutations affecting the expression of specific genes in immune cells can modulate their anti-tumor activity [18, 19]. Although these mutations do not directly "drive" tumor growth, they may modulate it by inhibiting or enhancing anti-tumor immune response.

Clonal hematopoiesis occurs in over 10% of adults over 70 years of age [20] and has been implicated in cardiovascular and hematologic disorders, including leukemias and lymphomas [21, 22]. Over 20 different mutations have been associated with clonal hematopoiesis, with mutations in *DNMT3A*, *TET2* and *ASXL1* accounting for over 90% of age-related cases [23]. Although the associated mechanism is poorly understood, studies suggest three possible mechanisms: increased self-renewal, increased number of self-renewal cycles to become a committed progenitor, and/or increased epigenetic or transcriptional heterogeneity leading to a highly proliferative state [24–27]. However, clonal hematopoiesis by itself may not affect immune function. Instead, secondary mutations in clonally expanding cells may be required for immune dysfunction [23, 28, 29], which could affect anti-tumor immune function.

We hypothesize that the presence of pathogenic CH mutations in TII cells could impact anti-tumor immune function and therefore, could serve as predictive markers for cancer risk. In the following, we develop a machine learning model to predict the probability of cancer based on the detection of these mutations in blood samples.

## Materials and methods

### Exome sequencing data and variant calling

For data from the Sequence Read Archive (SRA) [30], fastq files from the repository were aligned to the GRCh38 reference genome using the Burroughs-Wheeler aligner (BWA) v0.7.15 [31]. For data from the Genomic Data Commons (GDC), we downloaded aligned read (bam) files which were already aligned to the GRCh38 reference genome using BWA. We used the same aligner (BWA) and reference genome (GRCh38) for SRA, as was used for the GDC data to produce the bam files. Variants were then called using GATK Mutect2 v3.7.0–1 [32]. The same read alignment and variant calling protocols were used for all datasets. The Clara parabricks v4.0.0–1 [33, 34] implementation of these programs was used for parallel execution

on NVIDIA GPUs. Default values were used for all parameters, including minimum variant calling accuracy (Log Odds > 3.0). To identify potentially pathogenic mutations only, we also downloaded Mutect2 mutation annotation format (MAF) files from the GDC, which include variant allele fraction (VAF) values for matched tumor and blood samples. The MAF files were not used to train or test the ML model.

### Single-cell RNA sequencing (scRNA-seq) data and cell type analysis

Cellranger v6.0.2 [35] was used to identify genes expressed in each cell, from scRNA-seq fastq files. SCSA v1.1 [36] was used to identify cell types from k-means clustering of cells based on the genes expressed by each cell. Default values were used for all parameters.

### Training, validation, and test sample sets

Ten of 18 datasets were randomly split into Training, Validation, and Test sets using the train_test_split() module in the Python scikit learn v1.2.2 library [37]. The other eight datasets were used as the independent test set.

### Machine learning (ML) algorithms

The ML algorithms implemented in the scikit_learn v1.2.2 library were used to build the ML models [37]. See S3 Table in S1 File for hyperparameter values.

### Statistical analysis

The binomial proportion confidence interval (CI) with a normal approximation was used to calculate CI for accuracy, using the proportion_confint() module in the Python statsmodels v0.13.15 library [38]. Standard errors for Logistic Regression (LR) coefficients were estimated using the bootstrap method, using 1000 iterations of sampling with replacement [39]. The estimated standard error was used to calculate Z-statistic, CI, and p-value for the LR coefficients.

## Results

In this study we first identified a set of 98 potentially pathogenic (non-passenger) CH mutations in TII cells in lung cancer tissue samples, following the approach in Ref [40] and detailed below. The variant allele fraction (VAF) for these mutations in blood samples was then used as the feature set (predictors) for developing a machine learning model for differentiating between lung cancer and non-cancer cases. A logistic regression model based on these mutations in blood samples was able to differentiate between a test set of lung cancer and non-cancer samples with 90.26% accuracy (95% CI: 88.50–92.01%).

### Potentially pathogenic CH mutations in TII cells

Our approach for identifying potentially pathogenic CH mutations in TII cells consisted of three stages (Fig 1a). First, we identified protein altering variants in the TCGA dataset for lung adenocarcinoma (LUAD) [41]. These variants consisted of missense, nonsense, indel, and splice site mutations (Fig 1b). Other mutations, such as intergenic, intronic and synonymous mutations are less likely to be pathogenic and were therefore excluded. Of the 2.5 million distinct variants in the 569 TCGA-LUAD samples, 428,015 were protein-altering (Fig 1a).

From the above set of protein-altering mutations, we selected CH mutations in TII cells based on variant allele fraction (VAF) in matched tumor and normal blood samples. Clonal hematopoiesis is conventionally defined as somatic mutations with a VAF > 2% [20]. This lower limit excludes mutations in circulating tumor cells or cell-free DNA which have a

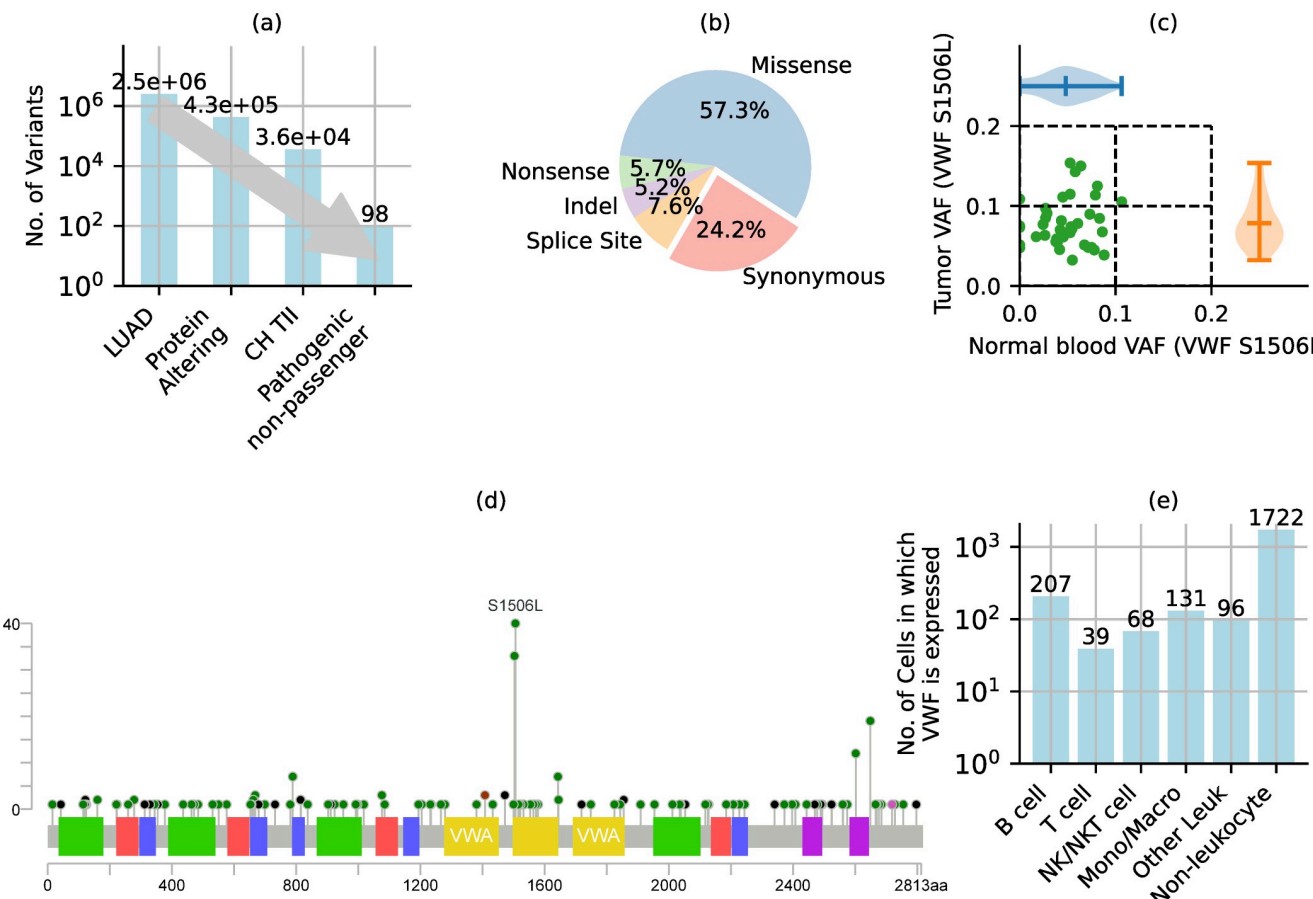

**Fig 1. Identifying potentially pathogenic clonal hematopoietic (CH) mutations in tumor infiltrating immune (TII) cells.** (a) Mutations were limited to protein-altering variants that were clonally expanded in TII cells and were considered to be potentially pathogenic non-passenger mutations. (b) Protein altering mutations excluded intergenic, synonymous, and intronic variants. (c)-(e) Example of criteria used to select the S1506L variant in VWF. (c) CH mutations in TII were identified by variant allele fraction (VAF) in tumor and normal blood samples. (d) One of the four criteria for identifying potentially pathogenic mutation was that the mutation occurred in > 5% of tumor samples, but rarely (< 0.01%) in the genome aggregation database (genomAD). (e) Another criterion used to identify potentially pathogenic mutations in TII was that the gene was expressed in tumor infiltrating immune cells based on lung cancer single-cell RNA sequencing data.

VAF < 1% [42, 43]. To exclude potential germline mutations, we selected variants with a VAF < 25%. To identify mutations in TII cells, we further selected variants that occurred in both tumor and matched blood samples. It is highly improbable that the same somatic variant would originate in tumor and blood cells simultaneously. It is more likely that these mutations originated in blood cells and subsequently infiltrated the tumor microenvironment, leading to their detection in tumor infiltrating blood cells. For example, the S1506L mutation in the gene for von Willebrand factor (*VWF*) occurs with a median VAF of 7.23% in TCGA-LUAD tumor samples containing the mutation, and in 4.90% of matched blood samples (Fig 1c). Of the 428,015 protein-altering variants, 36,293 were CH mutations in TII cells (Fig 1a).

From the above set of CH mutations in TII cells, we selected potentially pathogenic mutations based on the following three criteria. First, the mutation occurred frequently in tumor samples (> 5% of TCGA-LUAD samples), but rarely in large population-based studies (< 0.01% of sequences in the genome aggregation database (gnomAD)), suggesting that the genomic location is highly conserved in the germline, but somatic mutations may be correlated

with tumor progression. For example, the S1506L variant in *VWF* occurred in 6.85% of the 569 TCGA-LUAD samples (Fig 1d), whereas it was not detected in any of the gnomAD germline sequences (S2 Table in S1 File). Second, the gene was expressed in TII cells based on the analysis of single-cell RNA-sequencing data (Methods) for 87,743 cells from 21 lung tumor samples [44–46]. Genes not expressed in TII cells may not be functional in this context and were therefore excluded. For example, *VWF* was expressed in lung tumor infiltrating B, T, NKT, and monocyte/macrophage cells (Fig 1e). Lastly, the mutation was predicted to be damaging by two mutation-significance prediction tools–SIFT [47] and PolyPhen2 [48] (S2 Table in S1 File). Of the 36,293 protein-altering CH mutations in TII cells, 98 were potentially pathogenic mutations based on the above criteria (Fig 1a). The 98 potentially pathogenic CH mutations in TII cells are listed in S2 Table in S1 File, along with values for the criteria used to identify them. Although the selected mutations are unlikely to be passenger mutations, they are also not "driver" mutations, in the sense that they do not directly cause tumor cell growth. However, by potentially affecting immune cell function, they may modulate tumor growth and may therefore represent potential predictors of cancer risk.

## Machine learning (ML) study design

To develop an ML model with realistic prediction potential using available sequencing data, a key consideration was mitigating potential confounding or batch effects. In general, large-scale cancer DNA sequencing studies do not include non-cancer controls [41], and large-scale non-cancer studies do not include cancer cases [49]. Although we used the same protocol for aligning sequencing reads to the reference genome and for variant calling, these studies may use different protocols for cohort selection and sequencing. Therefore, ML models may classify cases based on differences in mutations detected that are artifacts of these protocol differences rather than characteristics related to the presence of cancer. To mitigate such confounding (batch) effects, we used data from multiple different studies. We also evaluated alternative models using the average batch accuracy to avoid bias toward protocols used by the larger studies. For example, two of the non-cancer cohorts, prjna532465 and prjna790003, had average ages of 82 and 77 years, respectively, significantly higher than the 63 years for the lung cancer samples (Table 1). However, by including the non-cancer cohorts, prjna421434 and prjna342304, with average ages of 42 and 39 years, respectively, the overall average age for the non-cancer samples was 64 years, comparable to the average age for lung cancer samples, to the extent the data is available. As noted in the Discussion section below, these strategies can mitigate the confounding effects but cannot fully eliminate them.

A set of whole exome sequencing data for blood samples from 18 different cohorts were identified for this study (Fig 2, Table 1). These included a total of 1,992 lung cancer and non-cancer blood samples. On average, for the cohorts for which demographic data was available, the age and gender composition for the lung cancer and non-cancer cohorts were comparable. The average age and gender composition were 63 years and 56% male for the lung cancer cases, respectively, and 60 years and 51% male for the non-cancer cases (Table 1). Although the racial compositions were significantly different with 71% white in the lung cancer cohorts and 42% white in the non-cancer cohorts, racial information was not available for nine of the 18 cohorts and two of the non-cancer cohorts targeted Asian populations. Although the percentage of current or past smokers in lung cancer cases (63%) was comparable to non-cancer cases (66%), smoking status was only available for two of the nine non-cancer cohorts (Table 1). The proportion of samples by stage also varied by cohort, with batch averages of 24.18% (0–54.35%) for stage I, 21.91% (0–40.74%) for stage II, 21.32% (0–64.71) for stage III, 20.60% (0–100%) for stage IV, and 11.93% (0–46.15%) for pre-cancerous cases (S1 Table in S1

**Table 1. Datasets used for model training, validation, and testing.**

| Study ID | Number of Samples | | | | Avg. Read Depth | Avg. Age (yrs.) | % Male | % White | % Past or Current Smokers |
|---|---|---|---|---|---|---|---|---|---|
| | Total | Training | Validation | Test | | | | | |
| Lung Cancer | | | | | | | | | |
| prjeb50602 [51] | 16 | 9 | 3 | 4 | 117 | 53 | 60% | NA | NA |
| prjna439205 [52] | 4 | 2 | 1 | 1 | 260 | 68 | 41% | NA | >62% |
| prjna610493 [53] | 32 | 18 | 6 | 8 | 167 | NA | 66% | NA | NA |
| prjna613408 [54] | 13 | 6 | 3 | 4 | 166 | 62 | 15% | NA | 0% |
| tcga-luad [55] | 440 | 247 | 83 | 110 | 177 | 65 | 46% | 86% | 81% |
| cptac-luad [56] | 113 | | | 113 | 341 | 62 | 65% | 31% | 58% |
| cptac-lusc [57] | 102 | | | 102 | 333 | 66 | 80% | 78% | 78% |
| prjeb47088 [58] | 22 | | | 22 | 36 | NA | NA | NA | NA |
| tcga-lusc [59] | 214 | | | 214 | 173 | 68 | 72% | 90% | 93% |
| **Total / Average** | **956** | **282** | **96** | **578** | **197** | **63** | **56%** | **71%** | **63%** |
| Non-cancer | | | | | | | | | |
| prjeb35045 [60] | 58 | 32 | 11 | 15 | 95 | NA | NA | 17% | NA |
| prjna167318 [61] | 163 | 91 | 31 | 41 | 170 | NA | 56% | NA | NA |
| prjna262923 [62] | 190 | 106 | 36 | 48 | 195 | NA | NA | NA | NA |
| prjna342304 [63] | 46 | 25 | 9 | 12 | 191 | 39 | 38% | 96% | NA |
| prjna790003 [64] | 200 | 112 | 38 | 50 | 166 | 77 | 49% | 0% | 67% |
| prjeb5736 [65] | 147 | | | 147 | 68 | NA | NA | 0% | NA |
| prjna421434 [66] | 91 | | | 91 | 324 | 42 | 76% | NA | 64% |
| prjna532465 [67] | 38 | | | 38 | 457 | 82 | 37% | 98% | NA |
| prjna59849 [49] | 103 | | | 103 | 118 | NA | NA | NA | NA |
| **Total / Average** | **1036** | **366** | **125** | **545** | **191** | **60** | **51%** | **42%** | **66%** |

For non-cancer cohorts, the demographic information is for the whole study even though we selected a subset of samples from each of the studies. Averages are not weighted by sample size. Unweighted averages are more relevant for the batch average optimization used here. For prjna167318, the selected samples consisted of the parents of autism patients. NA = Not Available.

File). Five of the nine lung cancer cohorts were all treatment naïve and two of the cohorts were cases with previous treatments (S1 Table in S1 File). Methods for accounting for batch-effects is an active area of research [50], however it is unclear how effective these methods would be for somatic mutation data, or if they would result in the loss of critical information.

All data from eight of the 18 cohorts were set aside as an Independent Test set, representing a total of 830 samples (Fig 2). Since there is no single large cohort containing both lung cancer and non-cancer sequencing data, the independent test set represents the best possible test of the predictive potential of the model developed below, short of a blinded clinical case-control study where we can proactively control all possible confounding variables. In addition, 25% of the data (293 samples) from each of the other 10 cohorts were also set aside for testing (Fig 2, Table 1). None of the samples in the Test set were used to develop the model. The remaining 75% of the data (869 samples) were further subdivided into a Training set (648 samples) and a Validation set (221 samples). Studies have shown that this sample size can be sufficient for >90% concordance in prediction accuracy, indicating low risk of overfitting [68–70]. The Training and Validation data sets were used to evaluate six commonly used ML algorithms: Light Gradient Boosted Trees, Random Forest, Support Vector Machine, Neural Network, k Nearest Neighbors, and Logistic Regression classifiers. Combinations of model hyperparameters were evaluated for each of these algorithms using the Training and Validation sets. In

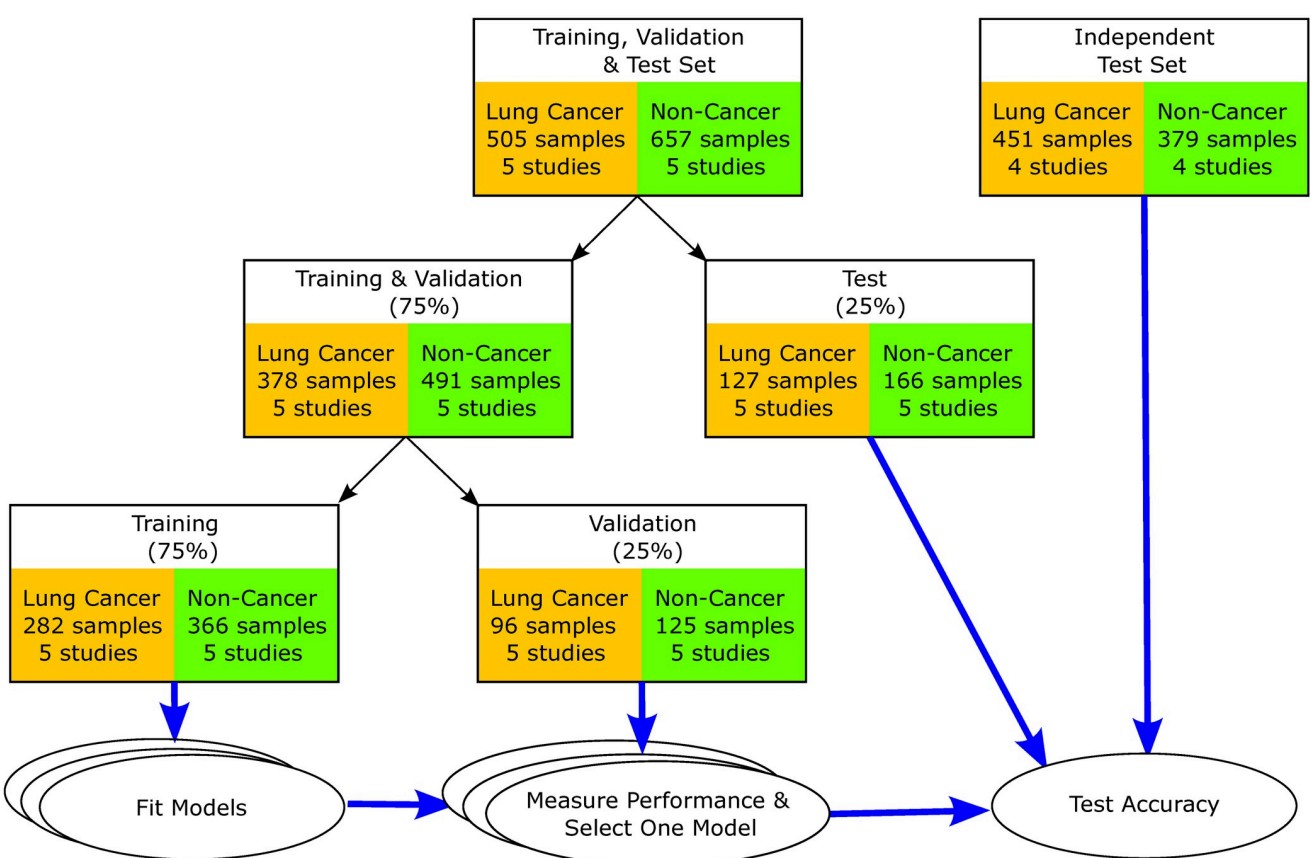

**Fig 2. Machine learning study design.** Sequencing data from multiple different cohorts were used to mitigate batch-effect artifacts due to differences in cohort selection and sequencing protocols. Training and Validation set: 75% of the sequencing data from 10 of 18 different cohorts were used to train the machine learning models and identify a model that had the highest average batch accuracy. Test set: All data for 8 of the 18 cohorts, in addition to 25% of the data from the other 10 cohorts.

addition to model hyperparameters, we also evaluated the effect of dichotomizing the VAF based on cutoff values ranging from 0.00 to 0.30. Each of the resulting 135,811 models was evaluated using the Validation set (Fig 2). Average unweighted batch accuracy was then used to identify the best model for each of the six ML algorithms. Average batch accuracies for all six models were comparable, ranging from 91.93% (66.67–100.00%) for Logistic Regression to 96.66% (83.33–100.00%) for Light Gradient Boosted Trees (S3 Table in S1 File, Fig 3a).

### Logistic regression (LR) model

Since average batch accuracies for all six ML models were comparable (91.93–96.66%), we selected the LR model for the direct real-world interpretability of its parameters. Specifically, the log odds-ratio (LOR) of model predictions and feature weights. The LR model can be formulated as

$$\ln \frac{p}{1-p} = \beta_0 + \sum_{i=1}^{n} \beta_i x_i \tag{1}$$

where $p$ is the predicted probability of cancer, $\frac{p}{1-p}$ is the odds-ratio for cancer prediction, $\beta_0$ is the intercept, $\beta_i$ are the weights for each of the features (predictors) $x_i$, and $n$ is the number of

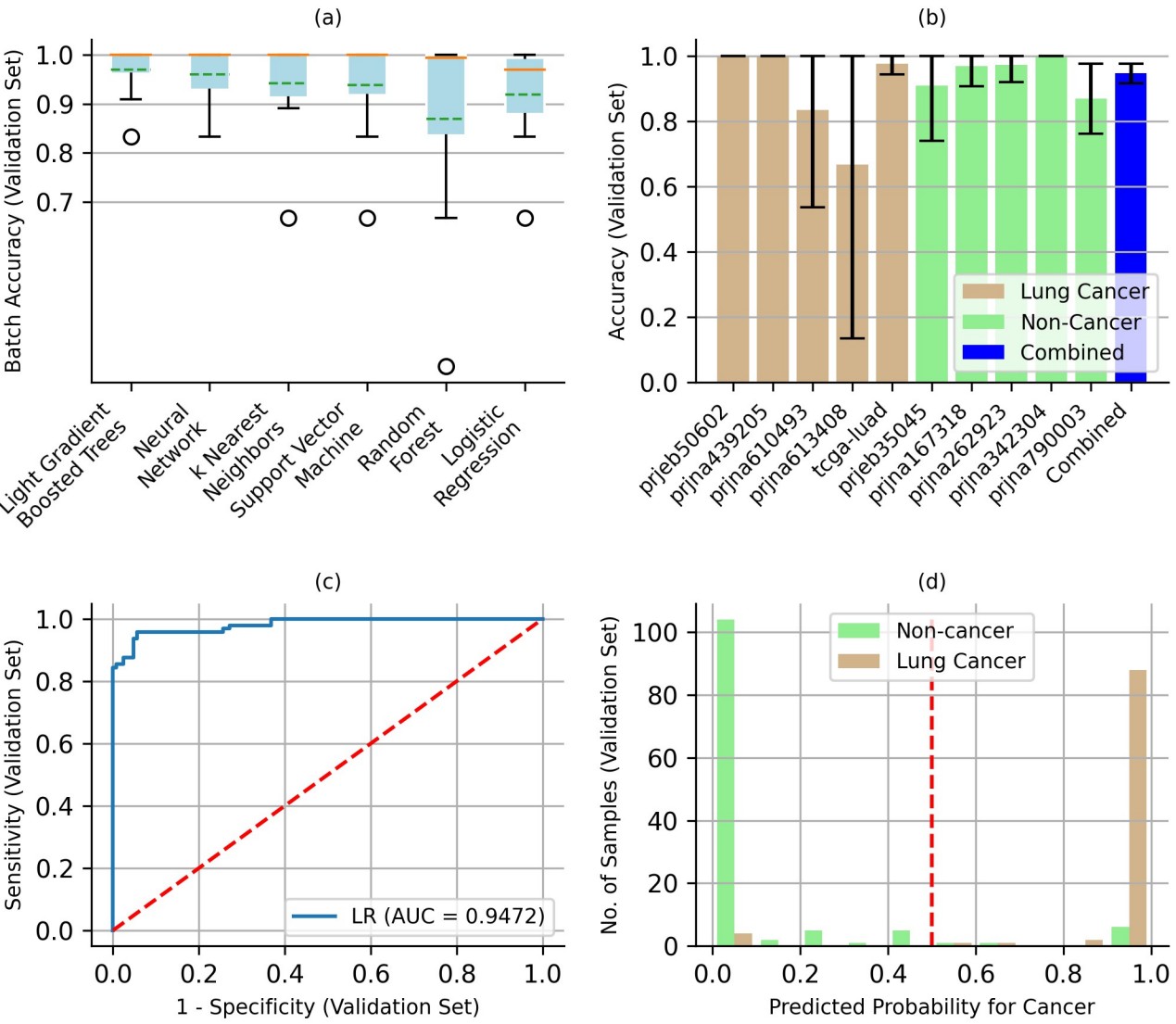

**Fig 3. Selecting an optimal machine learning classifier.** (a) Batch accuracy for the different machine learning algorithms, with model hyperparameters that produced the highest average batch accuracy for the validation set. Error bars show the range of values, orange lines the median and green dashed lines the mean. (b) A Logistic Regression (LR) model was selected for its interpretability of feature coefficients. Error bars show 95% confidence intervals. (c) Receiver Operating Characteristic (ROC) curve for the LR model showing 0.9472 Area Under the Curve (AUC). The red line represents a random prediction. (d) 91.66% of lung cancer and 83.20% non-cancer samples had predicted probabilities of >0.9 and <0.1 for cancer, respectively. Predicted probability of >0.5 (red line) predicts cancer and <0.5 no cancer.

features. $x_i$ = 1 or 0 corresponding to the presence or absence of the mutation with VAF > 0.01. All further testing was performed using the LR model. The weights $\beta_i$ for each of the predictors (mutations) can be interpreted as the LOR for the probability of cancer when *mutation = 1* versus *0*, controlling for all other predictors.

For the selected LR model, VAF was dichotomized as *mutation = 1* for *VAF>0.01* and *0* otherwise, *class_weight = {0:1, 1:4}*, *solver = 'saga'*, and default values for all other hyperparameters (S3 Table in S1 File). The accuracy of the LR predictions for each of the ten datasets in the Validation set ranged from a mean of 66.66% (95% CI: 13.32–100%) to 100% (Fig 3b). The overall accuracy for the combined Validation set was 94.57% (95% CI: 91.58–97.56%). For the

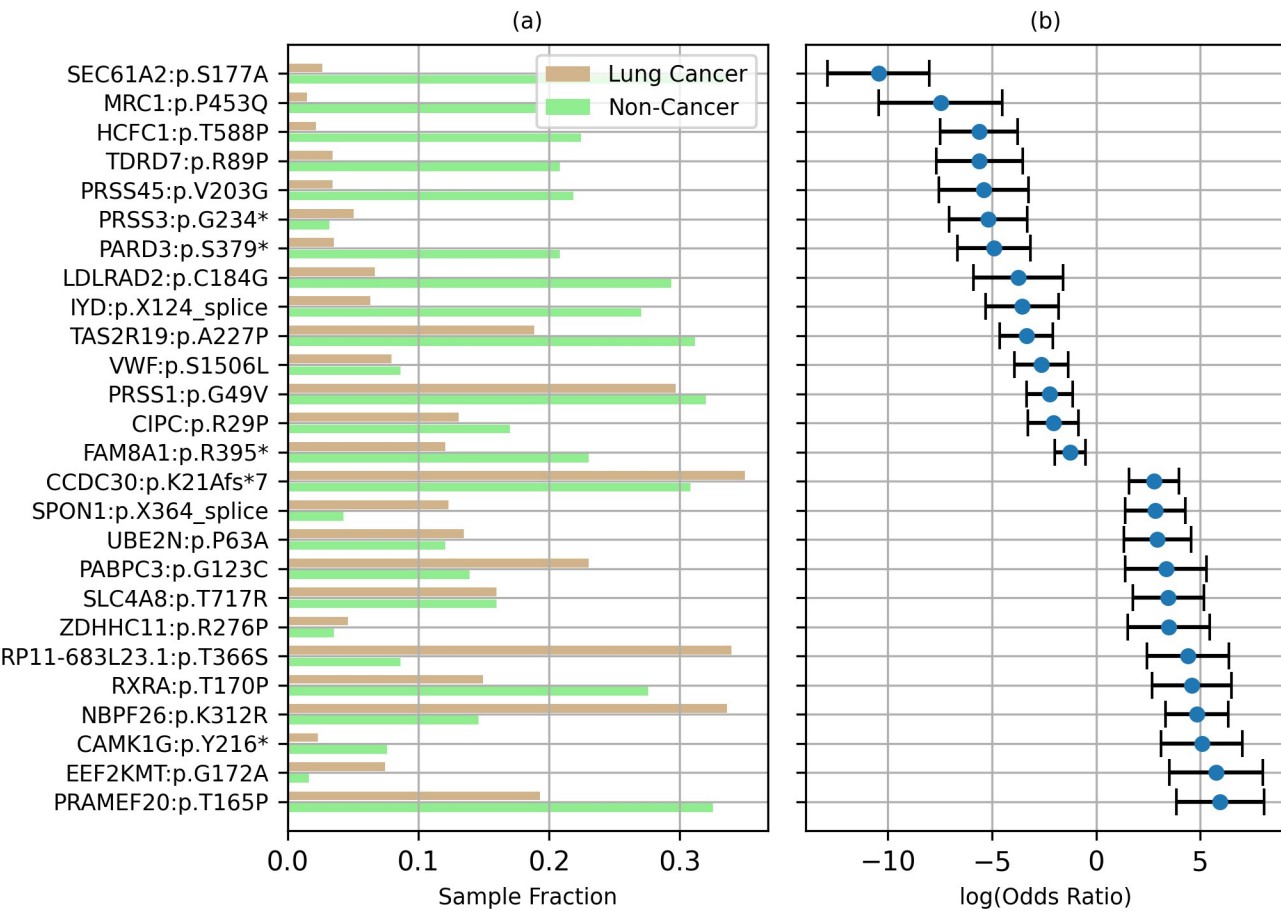

**Fig 4.** Mutations with p-value<0.05 for log odds ratio βi (a) Fraction of lung cancer and non-cancer samples with variant allele fraction (VAF) > 0.01. (b) Log odds ratio and 95% confidence interval.

Validation set, the receiver operating characteristic (ROC) curve had an area under the curve (AUC) of 0.9472, suggesting strong diagnostic ability (Fig 3c). With a cut-point of 0.5, sensitivity and specificity were 97.50% (95% CI: 94.71–100.00%) and 89.62% (95% CI: 83.82–95.43%), respectively. More importantly, 91.66% of the cancer cases were unambiguously predicted with probability of cancer > 0.90 and 83.20% of the non-cancer cases were predicted with probability of cancer < 0.10 (Fig 3d).

The LR model was refit to the combined Training and Validation set, to incorporate all available non-test data. The resulting model was then used for testing. For 87 of the 98 mutations the LOR $\beta_i \neq 0$ (Fig 4b), with the other 11 mutations ($\beta_i = 0$) not contributing to the LR model prediction. 47 of 87 mutations had $\beta_i > 0$, suggesting that these mutations may increase the risk of cancer, while 40 mutations had $\beta_i < 0$, suggesting a reduced risk, controlling for all other mutations. For nine of the 40 mutations with $\beta_i < 0$, the mutation occurred more frequently in non-cancer samples than in lung cancer samples. For 23 of the 47 mutations with $\beta_i > 0$ were found to occur more frequently in lung cancer samples (Fig 4a, S4 Table in S1 File). The LOR for 26 of the 98 mutations had p-value < 0.05 (Fig 4b, S4 Table in S1 File). Although the other 78 mutations individually have poor predictive power (p-value > 0.05), their combined effect can be highly predictive.

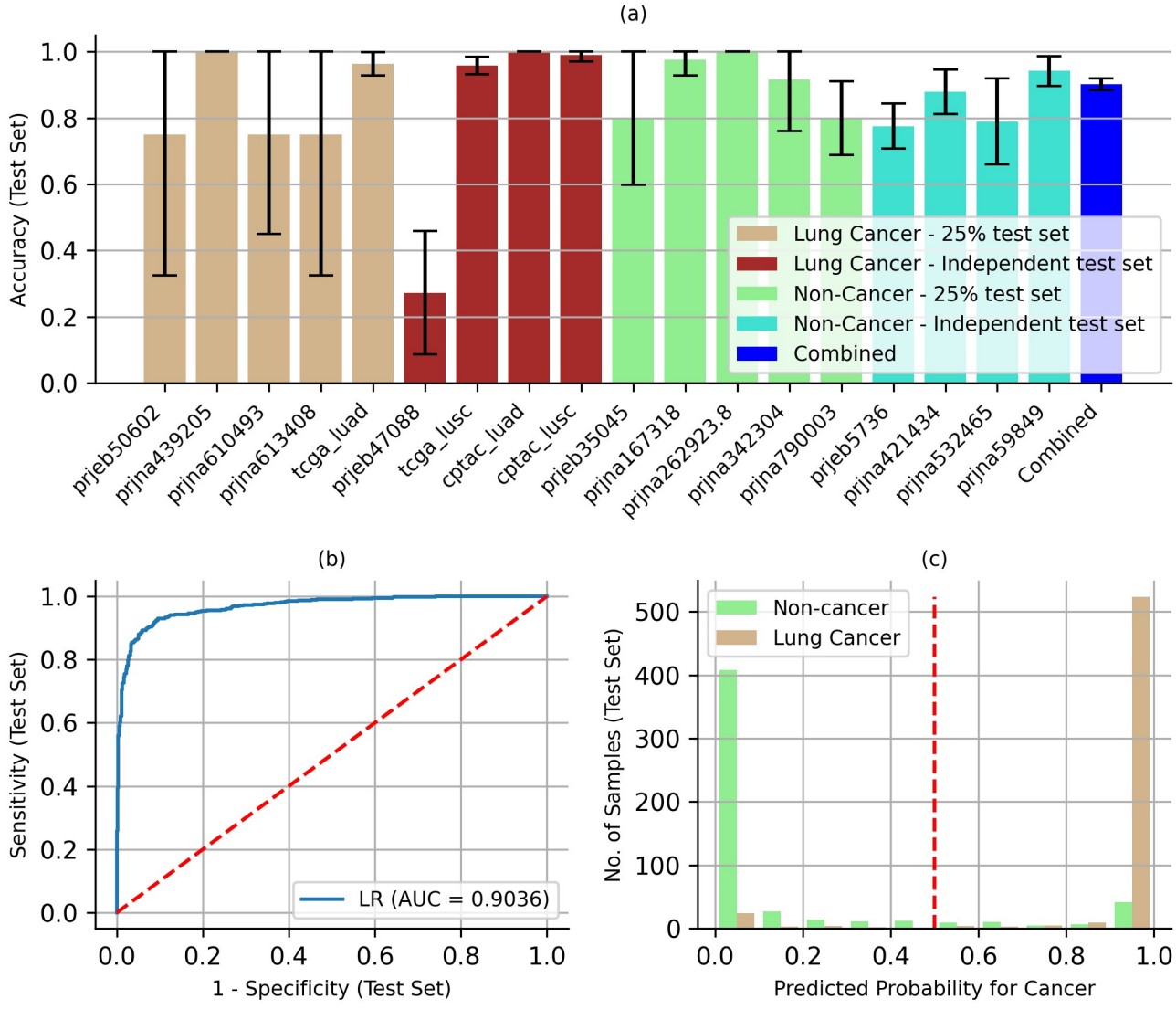

**Fig 5. Predictive accuracy for test datasets.** (a) Batch and Combined accuracy for the LR model. Error bars show 95% CI. (b) ROC curve for the LR model showing 0.9036 AUC. The red line represents a random prediction. (d) 90.48% of lung cancer and 74.86% non-cancer samples had predicted probabilities of >0.9 and <0.1 for cancer, respectively. Predicted probability of >0.5 (red line) predicts cancer and <0.5 no cancer.

## Test results

The Test set consisted of 25% of the data set aside from each of the ten studies used for Training and Validation (127 lung cancer and 166 non-cancer blood samples), and all the data from eight additional studies (451 lung cancer and 379 non-cancer blood samples) (Fig 2, Table 1). The average batch accuracy across all 18 datasets for the LR model developed above was 85.07% (27.27–100.00%) (Fig 5a). One of the 18 datasets, prjeb47088, had a very low accuracy, 27.27% (95% CI: 8.66–45.88%), due to a low read depth of 36 reads per exome loci containing the 98 mutations, compared to the average read depth of 197 across all cohorts (Table 1). With such a low read depth, mutations with low VAF (<5%) cannot be reliably detected [71], which can affect predictive accuracy. This finding highlights the potential confounding (batch) effect associated with sequencing depth. However, the relatively small number of samples (22) in this

dataset, accounting for only 1.96% of the 1,123 test samples (Table 1), did not significantly affect the overall accuracy of the model. The overall accuracy for the combined Test set was 90.25% (95% CI: 88.50–92.01%) (Fig 5a), comparable to the 94.57% accuracy for the combined Validation set (Fig 3b). In addition, the overall accuracy for the eight independent test cohorts alone was 89.88% (95% CI: 87.83–91.93%), excluding the 10 datasets where 25% of the data was set aside for testing. This was comparable to the 91.42% accuracy (95% CI: 88.06–94.77%) for the test data from other ten cohorts where 25% of the data was set aside for testing, suggesting that there was little if any confounding (batch) effect. The ROC curve had an AUC of 0.9036, suggesting strong diagnostic ability (Fig 5b). With a cut-point of 0.5, the sensitivity and specificity were 94.12% (95% CI: 92.20–96.04%) and 85.96% (95% CI: 82.98–88.95%). More importantly, 90.48% of the cancer cases were unambiguously predicted with probability of cancer > 0.90 and 74.86% of the non-cancer cases with probability of cancer < 0.10 (Fig 5c).

The contribution of each of the 98 mutations to the LOR of predicted probability of cancer for each sample is either 0 or $\beta_i$ (Eq 1). This contribution varies considerably from sample to sample (S1 Fig in S1 File). For some mutations the contribution is clear. The S177A mutation in *SEC61A2* with the most negative $\beta_i$ = -10.45 occurred in 55.96% of non-cancer samples but in only 5.61% of lung cancer samples (S1 Fig in S1 File). On the other hand, the T165P mutation in *PRAMEF20* with the largest $\beta_i$ = +5.95 is predictive of cancer, controlling for all other mutations. However, counter-intuitively, the mutation occurred in 57.43% of non-cancer samples and in only 28.05% of lung cancer samples. These results highlight the complex combination of weights and mutations in the LR model contributing to the predicted probability of cancer.

## Discussion

Differences in cohort selection and sequencing protocols, between lung cancer and non-cancer samples, can result in different variant profiles between the two sample sets. As a result, the Training and Validation stages may show artificially high accuracy but fail to accurately predict the probability on other independent datasets where the protocols may differ. To mitigate this confounding or batch effect we used datasets from ten different studies for training and validation, and eighteen different studies for testing. In addition, we used the average batch accuracy for the training and validation metric to mitigate the bias toward protocols for batches with larger number of samples. The comparable overall accuracy for the eight independent test cohorts alone (89.88%), excluding the datasets where 25% of the data was set aside for testing, and for the test data from other ten cohorts where 25% of the data was set aside for testing (91.42%), suggests that there was little if any confounding (batch) effect. Despite high test accuracy (Fig 5), these mitigation steps may not fully eliminate the batch-effect and the LR model developed here may not represent an optimal blood-based lung cancer screening panel. However, the LR model can serve as an effective starting point for developing an accurate lung cancer screening panel. To do so, a case-control study using a standardized protocol could be used to validate and optimize the LR model. The case-control study should also be more racially balanced than the studies used here, so that the screening panel is not racially biased.

The 98 potentially pathogenic mutations were selected because the associated gene sequences were highly conserved, the mutations were predicted to be damaging to gene function, and they could affect immune cell activity. Therefore, we expected most of the mutations to inhibit an effective anti-tumor immune response thus promoting tumor growth. However, many (48) of the 98 mutations had LOR < 0, suggesting that these mutations may be predictive of non-cancer cases. Chronic inflammation, triggered by tobacco smoke and other

carcinogens, has been shown to promote tumorigenesis [72, 73]. We speculate that these mutations may inhibit such chronic inflammatory responses possibly inhibiting tumorigenesis.

It is important to note that the presence of some CH mutations in peripheral blood may be due to the presence of cancer. Cancer is known to alter the peripheral blood immune cell composition, by triggering the expansion of specific immune cell subtypes [74]. These immune cell subtypes may harbor different mutations, which would then be more abundant in the altered peripheral blood due to clonal hematopoiesis (CH). The CH mutations identified by our algorithm may indeed represent the result of CH expansion of specific mutations in specific immune cell subtypes, rather than the general presence of these mutations in hematopoietic stem and progenitor cells. Further investigation will be required to clarify the relationship.

In addition to providing a starting point for developing a cancer screening panel, this study identified a set of potential treatment targets. The mutations selected as predictors for the LR model were based on their potentially pathogenic role in TII cells. The selection criteria are detailed in the Results section under "Potentially pathogenic CH mutations in TII cells". The values for the selection criteria for each mutation are listed in S2 Table in S1 File. Although the selection criteria used suggest the possibility that these mutations are pathogenic, we do not claim that that is necessarily so. While these mutations may not directly cause tumor growth, they may modulate it. In particular, the set of mutations with a p-value $< 0.05$ for the LOR (Fig 4), should be considered for further investigation. For example, the P453Q variant in *MRC1* had a LOR of -7.47 (95% CI: -2.52–-10.44), suggesting an inhibitory effect against lung cancer. *MRC1* codes for a membrane receptor protein that mediates macrophage endocytosis of glycoproteins [75]. *MRC1* expression has been associated with inflammatory macrophage phenotype [76]. We speculate that the mutation inhibits the expression of *MRC1*, possibly reducing chronic inflammation and inhibiting tumorigenesis. In contrast, the T170P variant in *RXRA* had a LOR of +4.60 (95% CI: 2.69–6.52), suggesting a facilitative effect. *RXRA* encodes a nuclear receptor that acts as a transcription factor promoting target gene expression. *RXRA* is known to inhibit non-small cell lung cancer cell growth [77] and is a therapeutic target for lung cancer [78]. The 26 mutations with significant LOR (p-value $< 0.05$, Fig 4, S4 Table in S1 File) may represent potential immunotherapy targets for lung cancer.

The lung cancer and non-cancer samples included in this study did not include samples from other cancer types. Therefore, a lung cancer screening panel based on the LR model may detect other cancer types as well. In fact, 36 of the 98 potentially pathogenic CH mutations in TII cells for lung cancer overlap with those for breast cancer identified in a previous study [40]. These mutations could result in a positive prediction in the presence of breast cancer. The approach presented here should be extended to develop a pan-cancer screening panel to differentiate between different cancer types.

## Conclusions

Early detection can significantly reduce the mortality rate due to lung cancer, yet the uptake of the currently approved low-dose computed tomography scan for lung cancer is limited. Here we present an approach for developing a blood-based screening panel that utilizes a set of potentially pathogenic clonal hematopoietic (CH) mutations detected in tumor infiltrating immune (TII) cells. While the effect of CH mutations in TII cells on solid tumor progression remains relatively unexplored, recent animal model studies suggest a potential role. We developed a logistic regression (LR) machine learning model for predicting the probability of lung cancer based on a set of 98 CH mutations in blood samples. The LR model demonstrated a high accuracy of 90.25% (95% CI: 88.50–92.01%) in a separate Test set. To mitigate batch-effects arising from differences in cohort selection and sequencing protocols, we used

sequencing data from 18 different studies, though this may not fully eliminate the batch-effect. A case-control study with standardized protocols could be used to validate and refine the LR model and develop an accurate blood-based lung cancer screening panel.

## Supporting information

**S1 File. The following tables and figures are included in the supporting information file.** S1 Table. Cohort characteristics. S2 Table. Potentially pathogenic CH mutations in TII. S3 Table. Best ML model hyperparameters and batch accuracy. S4 Table. LR model coefficients. S1 Fig. Average contribution of individual mutations to the prediction of cancer probability for cancer and control samples. (ZIP)

## Acknowledgments

We thank Drs. Robin T. Varghese and Harold Garner, VCOM, for their comments and suggestions.

## Author Contributions

**Conceptualization:** Ramu Anandakrishnan.

**Data curation:** Ramu Anandakrishnan.

**Formal analysis:** Ramu Anandakrishnan, Ryan Shahidi, Ian J. Zyvoloski.

**Funding acquisition:** Ramu Anandakrishnan.

**Investigation:** Ramu Anandakrishnan, Ryan Shahidi, Andrew Dai, Veneeth Antony, Ian J. Zyvoloski.

**Methodology:** Ramu Anandakrishnan, Ryan Shahidi.

**Project administration:** Ramu Anandakrishnan.

**Resources:** Ramu Anandakrishnan.

**Software:** Ramu Anandakrishnan, Ryan Shahidi, Andrew Dai, Veneeth Antony, Ian J. Zyvoloski.

**Supervision:** Ramu Anandakrishnan.

**Validation:** Ramu Anandakrishnan, Ryan Shahidi.

**Visualization:** Ramu Anandakrishnan, Ian J. Zyvoloski.

**Writing – original draft:** Ramu Anandakrishnan, Ryan Shahidi, Andrew Dai, Veneeth Antony, Ian J. Zyvoloski.

**Writing – review & editing:** Ramu Anandakrishnan, Ryan Shahidi, Andrew Dai, Veneeth Antony, Ian J. Zyvoloski.

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
