## [Decision Letter · Decision Letter 0]

28 May 2024

PONE-D-24-17330Blood-based screening panel for lung cancer based on clonal hematopoietic mutationsPLOS ONE

Dear Dr. Anandakrishnan,

Thank you for submitting your manuscript to PLOS ONE. After careful consideration, we feel that it has merit but does not fully meet PLOS ONE’s publication criteria as it currently stands. Therefore, we invite you to submit a revised version of the manuscript that addresses the points raised during the review process.  In particular, I agree with Reviewer #2: "... Authors must ... be consistent with the results by using a title that clearly indicates that a method is presented in this manuscript and not an already defined panel."Abstract, introduction and discussion sections must be clear about the limitations of the proposed approach.

We look forward to receiving your revised manuscript.

Kind regards,

Francesco Bertolini, MD, PhD

Academic Editor

PLOS ONE

Journal Requirements:

2. Please note that PLOS ONE has specific guidelines on code sharing for submissions in which author-generated code underpins the findings in the manuscript. In these cases, we expect all author-generated code to be made available without restrictions upon publication of the work. Please review our guidelines at https://journals.plos.org/plosone/s/materials-and-software-sharing#loc-sharing-code and ensure that your code is shared in a way that follows best practice and facilitates reproducibility and reuse."

   "This work was funded by the VCOM 2023 REAP grant #1038624(RA)."

Reviewers' comments:

Reviewer's Responses to Questions

**Comments to the Author**

1. Is the manuscript technically sound, and do the data support the conclusions?

Reviewer #1: Yes

Reviewer #2: No

2. Has the statistical analysis been performed appropriately and rigorously? 

Reviewer #1: Yes

Reviewer #2: Yes

3. Have the authors made all data underlying the findings in their manuscript fully available?

Reviewer #1: Yes

Reviewer #2: Yes

4. Is the manuscript presented in an intelligible fashion and written in standard English?

Reviewer #1: Yes

Reviewer #2: Yes

5. Review Comments to the Author

Reviewer #1: Interestingly, the authors show how mutations called on peripheral blood can raise a concrete signal useful to distinguish cancer vs healthy patients.

The test devised by the authors could indeed complement ctDNA or other diagnostic tests.

I wonder whether the presence of cancer in a patient could generate a peripheral blood mixture composed by a different percentage of cell types. Such different cells could harbour different mutations (or mutation rate) due to clonal hematopoiesis.

Major points:

1) It would be useful to encapsulate the test in a software application (with suitable instructions) and provide the possibility to download/run such applications on new datasets.

2) It would be interesting to know if you still get a signal completely separating the studies used for training vs tests (i.e. removing the 25% test set).

Reviewer #2: The authors want to determine whether clonal expansion of hematopoietic mutations in tumor-infiltrating immune cells in blood samples can predict lung cancer onset.

Using sequencing data from publicly available lung cancer samples, the authors identify a set of 98 potentially pathogenic clonal hematopoietic mutations in tumor-infiltrating immune cells. They then use these mutations as predictors to develop a logistic regression machine learning model. Finally, the authors test the results on a separate sequencing dataset of >500 lung cancer samples and >500 noncancer samples from 18 different cohorts.

The title suggests that the authors provide a panel of genes that can be used to predict the occurrence of lung cancer. The last sentence of the introduction (line 75) also suggests that a machine learning model is developed in this paper to predict the probability of cancer based on the detection of these mutations.

However, the results obtained are in contrast in that the authors write that the model developed here may not represent an optimal lung cancer screening panel (line 340) and in that they describe the work as a starting point that needs further development (line 355).

Authors must then be consistent with the results by using a title that clearly indicates that a method is presented in this manuscript and not an already defined panel. Consistency must be checked carefully throughout the manuscript.

At the methodological level, it is not convincing that different alignment and variant calling approaches are used depending on the dataset used. This approach is risky as it could introduce batch effects that may not be corrected by the methods used by the authors.

The authors should align reads and perform variant calling with one standardized approach for all datasets and re-run their method.

In addition, the authors make a number of functional assumptions (line 356) about the identified mutations (Fig. 4) but they do not validate or show data to support the assumptions made.

Additionally on line 349: 48/98 cannot be considered as “most,” since it is less than half.

6. PLOS authors have the option to publish the peer review history of their article (what does this mean?). If published, this will include your full peer review and any attached files.

Reviewer #1: No

Reviewer #2: No

---

## [Author Response · Author response to Decision Letter 0]

11 Jun 2024

Overall, we concur with reviewers' comments and suggestions.

Point-by-point response to reviewer comments

Reviewer 1 comment:

“The test devised by the authors could indeed complement ctDNA or other diagnostic tests. I wonder whether the presence of cancer in a patient could generate a peripheral blood mixture composed by a different percentage of cell types. Such different cells could harbour different mutations (or mutation rate) due to clonal hematopoiesis.”

Response:

This is an interesting question. Cancer is known to alter the peripheral blood immune cell composition, by triggering the expansion of specific immune cell subtypes. These immune cell subtypes may harbor different mutations, which would then be more abundant in the altered peripheral blood due to clonal hematopoiesis (CH). The CH mutations identified by our algorithm may indeed represent the result of CH expansion of specific mutations in specific immune cell subtypes, rather than the general presence of these mutations in hematopoietic stem and progenitor cells. Further investigation will be required to clarify the relationship. We have added a Discussion of this possibility in the manuscript. 

Reviewer 1 comment:

“1) It would be useful to encapsulate the test in a software application (with suitable instructions) and provide the possibility to download/run such applications on new datasets.”

Response:

We have packaged the code for predicting the probability of cancer from an input VCF files, with instructions for execution. It is available for download from: https://sourceforge.net/projects/lung-cancer-screening-panel/

Reviewer 1 comment:

“2) It would be interesting to know if you still get a signal completely separating the studies used for training vs tests (i.e. removing the 25% test set).”

Response:

The accuracy for the 8 independent test cohorts (89.88%) was comparable to the accuracy for the 10 cohorts where 25% of the data was set aside for testing (91.42%). These results are included in the Test Results subsection of the Results section.

Reviewer 2 comment:

“The title suggests that the authors provide a panel of genes that can be used to predict the occurrence of lung cancer. The last sentence of the introduction (line 75) also suggests that a machine learning model is developed in this paper to predict the probability of cancer based on the detection of these mutations. However, the results obtained are in contrast in that the authors write that the model developed here may not represent an optimal lung cancer screening panel (line 340) and in that they describe the work as a starting point that needs further development (line 355). Authors must then be consistent with the results by using a title that clearly indicates that a method is presented in this manuscript and not an already defined panel. Consistency must be checked carefully throughout the manuscript.”

Response:

We have updated the title to state that what we present here is “An approach for developing a blood-based screening panel for lung cancer based on clonal hematopoietic mutations”. We have also updated the manuscript throughout to indicate that we present an approach that needs to be further validated and optimized.

Reviewer 2 comment:

“At the methodological level, it is not convincing that different alignment and variant calling approaches are used depending on the dataset used. This approach is risky as it could introduce batch effects that may not be corrected by the methods used by the authors. The authors should align reads and perform variant calling with one standardized approach for all datasets and re-run their method.”

Response:

For the GDC datasets we used already aligned read files, whereas for the SRA read files we performed the alignment. However, we used the same alignment protocol for SRA dataset as was used for aligning the GDC datasets. Specifically, we used the Burroghs-Wheeler Aligner (BWA) with the GRCh38 reference genome for performing the alignment of SRA datasets, as was used to align the GDC datasets. The same variant calling protocol was then used for all aligned read files (Mutect2). In summary, the same approaches were used for sequence alignment and variant calling for all datasets. We have expanded the Methods section and updated other sections, to clarify.

Reviewer 2 comment:

“In addition, the authors make a number of functional assumptions (line 356) about the identified mutations (Fig. 4) but they do not validate or show data to support the assumptions made.”

Response:

On line 356 we stated that “The mutations selected as predictors for the LR model were based on their potentially pathogenic role.” The selection criteria are detailed in the Results section under the subsection “Potentially pathogenic CH mutations in TII cells”. The values for the selection criteria for each mutation are listed in Supplementary Table S2. The key word here is “potentially”. Although the selection criteria used suggest the possibility that these mutations are pathogenic, we do not claim that that is necessarily so. We only suggest that they warrant further investigation. We have expanded the text around line 356 to clarify. 

Reviewer 2 comment:

“Additionally on line 349: 48/98 cannot be considered as “most,” since it is less than half.”

Response:

Most replaced with Many. Thank you.

---

## [Decision Letter · Decision Letter 1]

2 Jul 2024

An approach for developing a blood-based screening panel for lung cancer based on clonal hematopoietic mutations

PONE-D-24-17330R1

Dear Dr. Anandakrishnan,

We’re pleased to inform you that your manuscript has been judged scientifically suitable for publication and will be formally accepted for publication once it meets all outstanding technical requirements.

Kind regards,

Francesco Bertolini, MD, PhD

Academic Editor

PLOS ONE

Additional Editor Comments (optional):

Reviewers' comments:

Reviewer's Responses to Questions

**Comments to the Author**

1. If the authors have adequately addressed your comments raised in a previous round of review and you feel that this manuscript is now acceptable for publication, you may indicate that here to bypass the “Comments to the Author” section, enter your conflict of interest statement in the “Confidential to Editor” section, and submit your "Accept" recommendation.

Reviewer #1: All comments have been addressed

Reviewer #2: All comments have been addressed

2. Is the manuscript technically sound, and do the data support the conclusions?

Reviewer #1: Yes

Reviewer #2: Yes

3. Has the statistical analysis been performed appropriately and rigorously? 

Reviewer #1: Yes

Reviewer #2: Yes

4. Have the authors made all data underlying the findings in their manuscript fully available?

Reviewer #1: Yes

Reviewer #2: Yes

5. Is the manuscript presented in an intelligible fashion and written in standard English?

Reviewer #1: Yes

Reviewer #2: Yes

6. Review Comments to the Author

Reviewer #1: (No Response)

Reviewer #2: All my previous comments have been addressed. The authors responded to my comments and made necessary changes to the title and manuscript.

7. PLOS authors have the option to publish the peer review history of their article (what does this mean?). If published, this will include your full peer review and any attached files.

Reviewer #1: **Yes: **Andrea Franceschini

Reviewer #2: No

---

## [Editor Report · Acceptance letter]

8 Jul 2024

PONE-D-24-17330R1 

PLOS ONE

Dear Dr. Anandakrishnan, 

I'm pleased to inform you that your manuscript has been deemed suitable for publication in PLOS ONE. Congratulations! Your manuscript is now being handed over to our production team.

Kind regards, 

on behalf of

Dr. Francesco Bertolini 

Academic Editor

PLOS ONE